# Relationship between Moral Values for Driving Behavior and Brain Activity: An NIRS Study

**DOI:** 10.3390/healthcare10112221

**Published:** 2022-11-07

**Authors:** Kaori Kawabata, Kazuki Fujita, Mamiko Sato, Koji Hayashi, Yasutaka Kobayashi

**Affiliations:** 1Graduate School of Health Science, Fukui Health Science University, Fukui 910-3190, Japan; 2Fukui Higher Brain Dysfunction Support Center, Fukui 910-0067, Japan; 3Department of Rehabilitation Medicine, Fukui General Hospital, Fukui 910-8561, Japan

**Keywords:** driving behavior, moral value, healthy driver, brain activity, near-infrared spectroscopy

## Abstract

Although there are clear moral components to traffic violations and risky and aggressive driving behavior, few studies have examined the relationship between moral values and risky driving. This study aimed to examine the relationship between moral views of driving behavior and brain activity. Twenty healthy drivers participated in this study. A questionnaire regarding their moral values concerning driving behavior was administered to the participants. Brain activity was measured using near-infrared spectroscopy while eliciting moral emotions. Based on the results of the questionnaire, the participants were divided into two groups: one with high moral values and the other with low moral values. Brain activity was statistically compared between the two groups. Both groups had significantly lower activity in the prefrontal cortex during the self-risky driving task. The low moral group had significantly lower activity in the left dorsolateral prefrontal cortex than the high moral group, while it had lower activity in the right dorsolateral prefrontal cortex in the self-risky driving task than in the safe driving task. Regardless of their moral values, the participants were less susceptible to moral emotions during risky driving. Furthermore, our findings suggest that drivers with lower moral values may be even less susceptible to moral emotions.

## 1. Introduction

Road traffic injuries are a major global health and development problem [1]. According to the 2018 Global Status Report on Road Safety, the annual number of deaths in road accidents has increased by approximately 100,000 in the last 3 years, and road accidents are now the leading cause of death among children and young people aged 5–29 years [2,3]. It has been reported that approximately 40–95% of road traffic injuries are a result of risky driving behavior [4,5,6]. Risky driving is an important cause of traffic casualties and a major health and cost problem for society [7,8]. In a traffic society, drivers have the responsibility and duty to obey traffic norms, pay attention and respect to all road users, and drive safely according to traffic conditions. However, there are drivers who, for one reason or another, engage in self-centered and risky driving behavior and ignore traffic norms [1,7,8]. In addition, they may also become angry at the words and actions of others, resulting in aggressive and risky driving behavior [9,10,11]. Traffic violations and risky or aggressive driving behaviors have a clear moral component, as they can cause great harm to oneself, others, and society as a whole [12].

Moral emotions play an important role in maintaining social norms and promoting prosocial behavior [13,14]. Weiner [15] broadly classified moral emotion into two categories: self-directed and other-directed. Self-directed emotions are directed toward oneself from the perspective of others, including negative emotions such as guilt and shame [16], which involve ongoing assessments of the moral worth and fit of the individual self within a community [17]. Other-directed emotions are directed from the self to others, including negative emotions such as anger. They reflect concern for the integrity of the social order, and recognition of a moral transgression often evokes feelings of contempt, anger, disgust, etc. [16].

Many studies have long shown that driver personality-related factors (emotions, personality, etc.) influence risky driving behavior [9,13,18,19,20,21,22]. Regarding the relationship between drivers’ moral emotions and risky driving behavior, negative other-directed emotions such as anger, distrust, and disgust while driving have been found to have a significant positive correlation with risky driving behavior [10,23]. Furthermore, a systematic review of the relationship between moral norms and risky driving behavior found that drivers with a high moral sense of obligation will abide by traffic rules and regulations, while drivers with a low moral sense of obligation are reluctant to abide by traffic rules and regulations [7]. Additionally, drivers with a low sense of legal obligation are believed to disregard traffic rules and have no guilt [7], and one of the factors contributing to dangerous and aggressive driving is a lack of norms and morality [9,22,24]. In other words, drivers with a low sense of moral obligation are less likely to generate negative self-directed emotions such as guilt and shame. In this report, a moral norm is defined as an individual’s obligation to perform the right behavior and avoid the wrong behavior, and the term “sense of moral obligation” is used as an inclusive expression, combining moral emotion and moral judgment.

Extensive research has been conducted to elucidate the neural basis of moral emotions, as they have a significant impact on daily life and social norms [25]. In particular, brain imaging studies using functional magnetic resonance imaging (fMRI, hereafter) have been widely conducted, suggesting that the frontal lobes play a major role [26,27,28,29]. However, fMRI requires the subject to lie supine in a narrow cylinder; moreover, the body, especially the head, must be firmly fixed, and the measurement time is long. In recent years, research on moral emotions using functional near-infrared spectroscopy (fNIRS) has attracted much attention and demonstrated its effectiveness as an alternative to fMRI [25,30,31,32], but there have been no reports focusing on drivers’ moral emotions or driving behavior. Furthermore, it is believed that judgments based on moral emotions are influenced by not only innate aspects but also acquired aspects, and despite the fact that different individuals have different moral views [33,34,35,36,37], only a few studies have examined brain activity in terms of individual differences in morality [25].

Considering the hypothesis that drivers with low morality may be less likely to generate moral emotions during hazardous driving than drivers with high morality, the research question for this study was set as follows: “Do different moral values of drivers lead to different activity in brain areas involved in moral emotions during risky driving”? Accordingly, the aim of this study was to examine the relationship between moral values for driving behavior and brain activity using NIRS. In this study, moral values were defined as “the idea of making judgments of right and wrong based on personal values and moral emotions”.

The novelty of this study is that it focused on individual differences in drivers’ moral values and examined brain activity during a moral judgment task simulating driving behavior. The findings will increase the understanding of neurological aspects of the relationship between drivers’ moral values and driving behavior. Furthermore, because inappropriate, immoral behavior is more likely to occur after brain injury [38,39], we believe that the findings will improve our understanding of the neural basis of immoral behavior in patients with moral–emotional disorders following brain injury and facilitate supportive measures to help them resume motor vehicle driving.

## 2. Materials and Methods

The study protocol was approved by the Ethics Review Committee of Nittazuka Medical Welfare Center (NE: 2022-13).

### 2.1. Participants

The participants were recruited between April and July 2022 in an automobile-dependent area of Japan. The inclusion criteria were (1) driving on a daily basis and (2) at least one year after obtaining a regular automobile license; the exclusion criteria were (1) cognitive decline or (2) a history of central nervous system disease. The participants were informed of the experiment in writing and orally, and informed consent was obtained from all participants. The participants were informed that the purpose of the study was to investigate the relationship between driving behavior and brain activity, and they were not told that the study would investigate their moral views regarding driving behavior in order to avoid socially desirable responses. There were 20 participants (12 males and 8 females) with a mean age of 25.3 years (SD = 6.8) and a mean driving experience of 6.9 years (SD = 7.0).

### 2.2. Questionnaire on Moral Values for Driving Behavior

First, a questionnaire was administered to all participants regarding their moral values for driving behavior. The questionnaire consisted of 15 items, which were translated into Japanese and partially modified based on aggressive driving behavior reported by Houston [10] (Table 1). The questionnaire had two versions: a self-version (regarding one’s own risky driving behavior) and an others-version (regarding the risky driving behavior of others). Participants answered the questionnaire in both versions, rating each item as good or bad on a scale from 0 to 5. A score of 0 was considered no problem (not bad), and a score of 5 was considered very problematic (very bad). Higher scores indicated higher moral values, and lower scores indicated lower moral values.

The median and mean scores for the 15 items were calculated. The median score for the self-version was 48.5, and that for the others-version was 48. Those participants with a score above the median of the self-version were defined as the group with high moral values (hereinafter, the high moral group), while those below the median score were defined as the group with low moral values (hereinafter, the low moral group). Two participants scoring within the median score range were excluded from the analysis. As a result, there were nine participants (five males and four females in both groups, respectively). The basic attributes of the participants in each group are presented in Table 2.

### 2.3. Research Procedure

The testing room was a quiet, private room where NIRS measurements were performed in the absence of sunlight. Brain activity was measured using NIRS (SMARTNIRS; Shimadzu Corporation, Kyoto, Japan) while presenting sentences concerning driving behavior to elicit moral emotions. Activity in the frontal lobe, which is associated with the generation of moral emotions, was measured [25,26,27]. The attachment area was identified according to the International 10–20 method, and the probe holder was placed on the frontal lobe so that the light-receiving probe 6 was positioned at the frontal pole (FPZ). The spacing between probes was 3 cm, and a total of 21 probes, seven horizontal rows × three vertical rows, for a total of 32 channels, were placed. The position of each probe was measured using a 3D position measuring device (FASTRAK, Polhemus Company, Yokohama, Japan).

Three tasks were performed: (i) safe driving, (ii) self-risky driving, and (iii) others-risky driving. (i) The safe driving task used sentences that complied with traffic rules. (ii) The self-risky driving and (iii) others-risky driving tasks used sentences described in the questionnaire on moral values for driving behavior. The protocol was measured in five consecutive sections, with 20 s of pre-task rest, 30 s of task, and 20 s of post-task rest as one block. At 30 s of the task, three sentences were presented for 10 s each. The three tasks were randomly ordered. The experimental design is shown in Figure 1a,b. The posture during the measurement was a chair-sitting position with plantar installation, and the participants were instructed to look at the screen of a laptop computer placed on a desk. Both hands were placed in a predetermined position on the desk, and verbal instructions were provided to constrain body movements and vocalizations during the measurement. Before the measurement began, the participants were instructed to silently read the sentences presented and make a judgment of right and wrong. In addition, the following instructions were added to the self-risky driving task: “Think about how much guilt or shame you would feel if you were to engage in any of these driving behaviors”. For the others-risky driving task, the following instructions were added: “Think about how uncomfortable or angry you would feel if you saw others engaging in these driving behaviors or if they did them to you”.

### 2.4. Processing of NIRS Data and Selection of Areas of Interest

Location estimation was performed using Fusion imaging software (Shimadzu Corporation) based on the probe location information obtained with FASTRAK. Then, the cortex localization of the channels was obtained using Statistical Parametric Mapping for Near Infrared Spectroscopy (NIRS-SPM), which was programmed with MATLAB R2014a (MathWorks Inc., Natick, America). The dorsolateral prefrontal cortex (DLPFC) and prefrontal cortex (PFC) are regions of interest that are particularly involved in moral emotions [26,31]. A total of 24 channels (Ch) in these regions of interest were included in the analysis (Figure 1a).

Noise in the NIRS signal was corrected using a 0.01–0.2 Hz band-pass filter and independent component analysis [40]. Among oxygenated hemoglobin (Oxy-Hb), deoxygenated hemoglobin, and total hemoglobin, we focused on Oxy-Hb, which is considered to reflect neural activity most sensitively, and analyzed it in terms of brain activity. The 5 s of rest immediately before the start of the task were used as the baseline, the 20 s of data starting from 10 s after the start of the task to the end of the task was normalized by the standard deviation of the baseline, and the Z-score was calculated representing the change in Oxy-Hb.

### 2.5. Statistical Analysis

Statistical comparisons were made for age, driving history, moral values for driving behavior, and Oxy-Hb changes between the two groups. Age and driving history were analyzed using the Mann–Whitney U test. The self-version and others-version scores of moral values for risky driving between the groups were analyzed using repeated-measures analysis of variance (ANOVA; GROUP × VERSION). The amount of change in Oxy-Hb in the three tasks between the groups was analyzed by repeated-measures analysis of variance (ANOVA; GROUP × TASK). Multiple comparison tests (Bonferroni) were performed as post-tests. All statistical analyses were performed using EZR (Saitama Medical Center, Jichi Medical University, Saitama, Japan) [41], and the significance threshold was set at *p*-value < 5%.

## 3. Results

The mean age and mean driving history of the high moral group/low moral group were 23.9 years (SD = 5.7) and 5.4 years (SD = 6.0)/25.9 years (SD = 7.0) and 7.4 years (SD = 7.2), with no significant differences between the two groups (Table 2).

The mean scores of moral values for driving behaviors (self-version and others-version) for the high moral group/low moral group were 62.6 (SD = 6.9) and 55.1 (SD = 11.2)/41.4 (SD = 3.0) and 41.2 (SD = 8.0). There was a significant main effect of GROUP (F(1,14) = 39.130, *p* < 0.001), but no significant main effect of VERSION or interaction between GROUP and VERSION (Figure 2).

The changes in Oxy-Hb levels in the three tasks in both groups are shown in Figure 3a,b. There was a significant interaction effect between GROUP and TASK in Ch14 and Ch27 (F(2,12) = 3.927, *p* = 0.029, F(2,12) = 4.966, *p* = 0.013, respectively). Post-test results showed no significant difference for Ch14 but for Ch27, the low moral group showed significantly less brain activity in the self-risky driving task than in the safe driving task (*p* = 0.040). In Ch12, there was a significant main effect of GROUP (F(1,12) = 4.765, *p* = 0.044), indicating that brain activity was significantly lower in the low moral group than in the high moral group. In Ch15 and Ch30, there was a significant difference in the main effect of the task (F(2,12) = 4.185, *p* = 0.024 and F(2,12) = 4.581, *p* = 0.017, respectively), with brain activity being significantly lower in the self-risky driving task than in the safe driving task (*p* = 0.003, *p* = 0.014, respectively).

That is, the low moral group had less activity in the area corresponding to the left DLPFC than the high moral group, and the low moral group had less activity in the area corresponding to the right DLPFC in the self-risky driving task than in the safe driving task. In addition, the left and right PFC were less active in the self-risky driving task than in the safe driving task, regardless of the moral value.

## 4. Discussion

In this study, we investigated the moral view of driving behavior in healthy drivers and measured their brain activity using NIRS while eliciting moral emotions. Two groups, one with high moral values and the other with low moral values, were created based on a questionnaire, and differences in brain activity were compared between these groups. Based on the characteristics of the traffic society, the present study focused on not only self-directed emotions such as shame and guilt but also other-oriented emotions such as anger and disgust as moral emotions toward risky driving. In the NIRS measurement task, the safe driving task was set as a control condition, in addition to the self-risky and others-risky driving tasks. Self-directed emotions were targeted in the self-risky driving task, and other-directed emotions were targeted in the others-risky driving task. The safe driving task targeted moral emotions during compliance with the traffic norms. In this case, positive emotions such as conscience and a sense of responsibility were considered to be affected, rather than negative emotions such as shame, guilt, anger, and disgust.

First, no significant differences were found in the main effects and interactions of the conditions on moral attitudes toward driving behavior, suggesting that the low moral group was less likely than the high-moral group to have negative feelings of shame, guilt, anger, and disgust toward risky driving, whether by themselves or with others. This is consistent with previous reports [5] that drivers with a low sense of moral obligation disregard traffic laws and do not feel guilty.

Next, brain activity was measured while eliciting moral emotions, and the results showed significant differences in the interaction and group main effects in the DLPFC. The DLPFC is generally considered to act in an inhibitory manner on emotional expression [42], and Paquette et al. [43] reported that brain activity in the same area increases when unpleasant emotions are elicited. The low moral group showed lower activity in the left DLPFC than the high moral group; this suggested that not only negative emotions, such as shame, guilt, anger, and disgust but also positive emotions, such as conscience and responsibility, were less likely to be induced. These results were generally consistent with our hypothesis. Furthermore, the low moral group had significantly less activity in the right DLPFC during the self-risky driving task, suggesting that they may be less likely to induce feelings of shame and guilt when they are driving risky. Since Mah et al. [44] stated that the DLPFC is related to the “theory of mind”, in which one infers the feelings and thoughts of others, it is possible that the low moral group has difficulty inferring how others feel when they drive risky. This could have a negative impact on others, not only because drivers from the low moral group might not hesitate to drive risky but also because they drive in a self-centered manner that lacks consideration for others. Drivers who are angry or disgusted with their driving environment have been reported to more frequently drive in a hostile or aggressive manner [10]; therefore, self-centered driving may contribute to problems with others.

Next, with regard to the PFC, we found significant differences in the main effects of the tasks. The PFC, including the medial PFC, was associated with emotional responses [45]. Anderson et al. [38,46] reported that when the prefrontal cortex is impaired, individuals are more likely to justify their actions by exhibiting emotional deficits such as guilt, remorse, and empathy. Both groups also showed lower activity in the areas corresponding to the left and right PFC in the self-risky driving task. Regardless of one’s moral views, when driving risky, one may be less susceptible to feelings of guilt or shame and more likely to justify one’s actions. However, the relationship between the DLPFC and PFC could not be analyzed in this study because NIRS cannot be used for connectivity analysis to calculate correlations between brain regions.

In addition, drivers with low moral values may be even less susceptible to moral emotions when driving, which may be more pronounced when they are driving riskily. In other words, they may be more likely to drive risky and may be a potential problem for others.

The present analysis of the relationship between moral values and brain activity with respect to driving behavior suggests that the amount of specific brain activity differs depending on differences in moral values and driving conditions, although it is difficult to say that there are marked differences in brain activity. van den Berg et al. [6] noted that the relationship between moral values and aggressive, risky driving behavior is uncertain, while at the same time noting that few reports have studied the relationship between moral values and risky driving. Moral emotions are involved in moral judgments, but their influence on the generation of judgments and judgment-based behavior is weak, and they go through several processing steps that are influenced by personal and situational factors [8,47]; therefore, further research is needed, including multivariate analyses that include the influence of other factors.

## 5. Limitations

The first limitation of this study was the small number of participants. Furthermore, the drivers who participated in this study were able-bodied individuals living in one automobile-dependent area of Japan. Moral values are influenced by acquired aspects such as environment and experience, which may lead to different results depending on the country and region in which one lives. For the generalization of the study findings, it is imperative to increase the number of participants and expand the areas covered by the survey. Furthermore, we believe that a comparison of healthy individuals and individuals with brain injury is necessary in order to develop supportive measures for the resumption of driving after brain injury. Finally, instead of presenting sentences that simulate driving behavior and induce moral emotions, researchers will need to clarify the relationship between actual driving behavior and brain activity.

## 6. Conclusions

We investigated the relationship between moral values and brain activity for driving behavior in healthy drivers using NIRS. Brain activity in two groups, one with high moral values and the other with low moral values, was compared. The results suggest that regardless of the difference in moral values, it is difficult to induce moral feelings during risky driving. In addition, drivers with low moral values may be less susceptible to moral emotions when driving, which may be more pronounced during risky driving.

## Figures and Tables

**Figure 1 healthcare-10-02221-f001:**
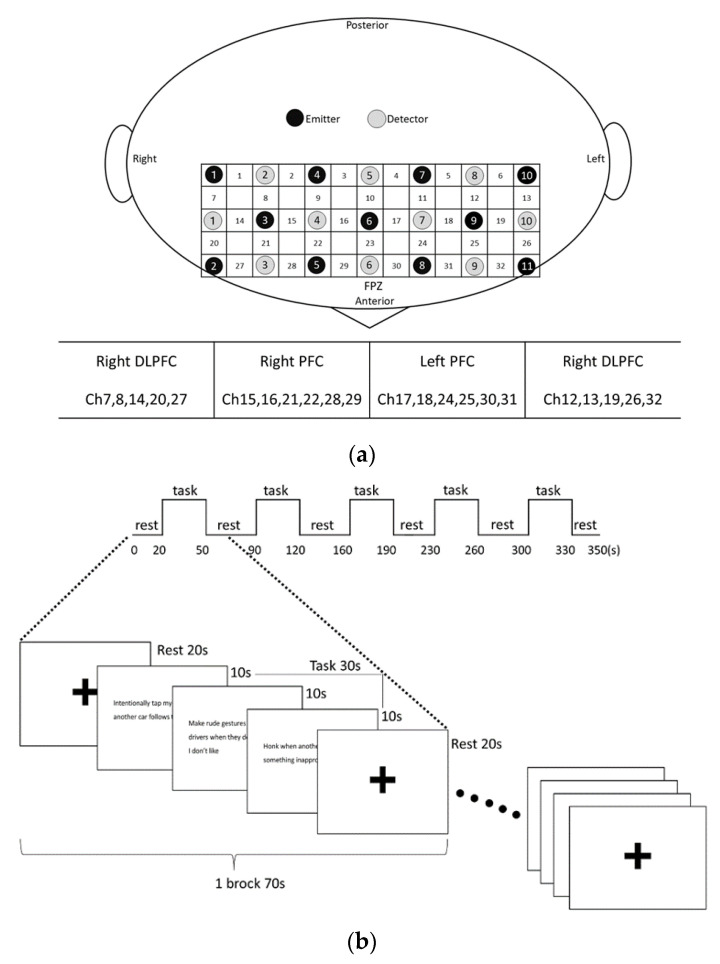
Experimental design. (**a**) Channel arrangement and areas of interest (DLPFC; Dorsolateral prefrontal cortex, PFC; Prefrontal cortex); (**b**) protocol and task design.

**Figure 2 healthcare-10-02221-f002:**
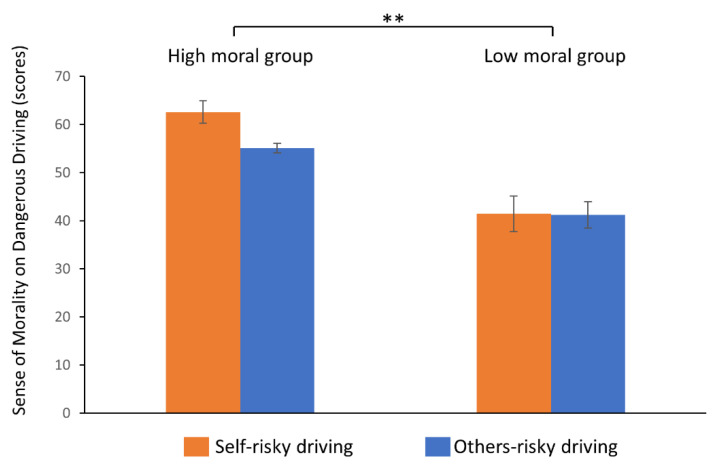
Results of moral values for driving behavior. Bars depict mean ± standard error. Statistics: Repeated measure two-factor ANOVA was performed; ** *p* < 0.01.

**Figure 3 healthcare-10-02221-f003:**
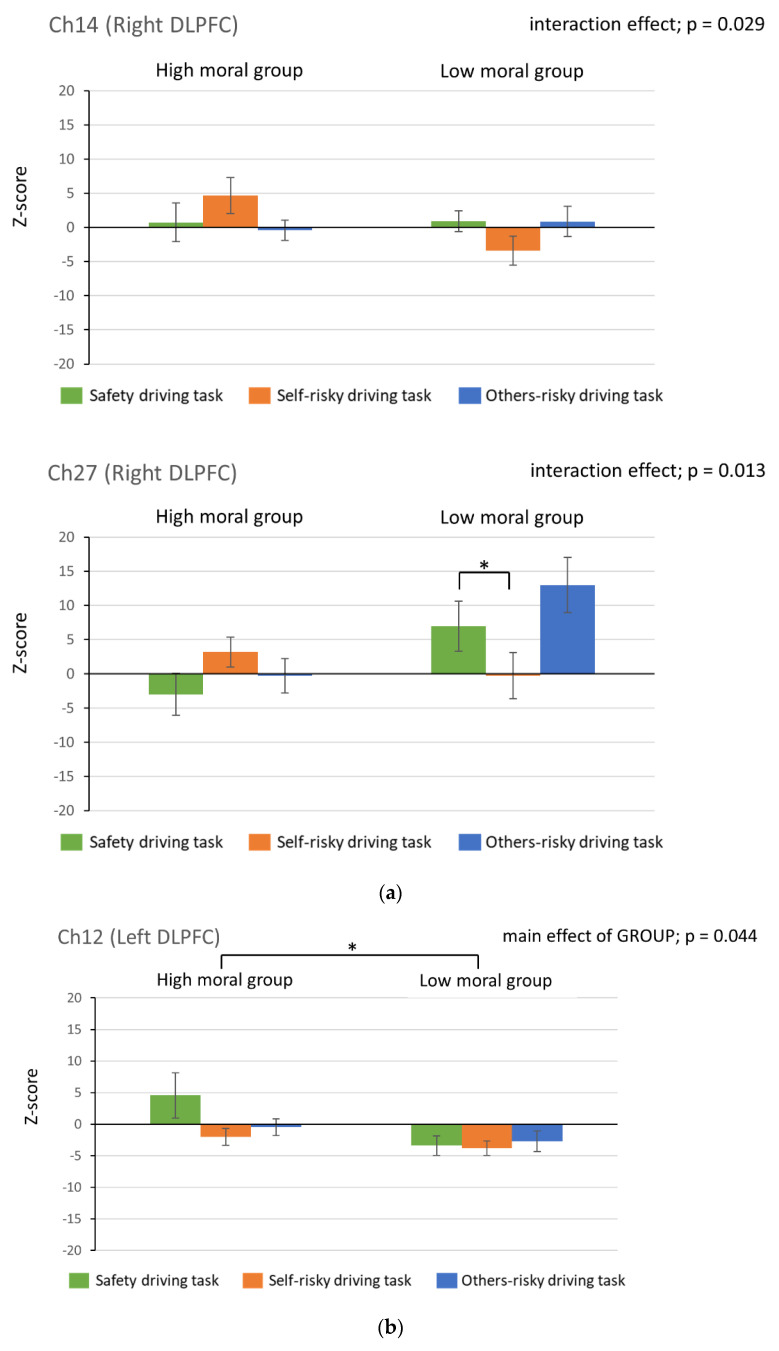
Results of brain activity by NIRS. (**a**) Channels that showed an interaction effect; (**b**) channels with a main effect of group; (**c**) channels with a main effect of task. The bars represent the mean ± standard error. In (**c**), the left side of the bar graph shows the high moral group, and the right side shows the low moral group. Statistics: repeated-measures two-factor ANOVA and Bonferroni correction were performed. * *p* < 0.05, ** *p* < 0.01.

**Table 1 healthcare-10-02221-t001:** Questionnaire on moral values for driving behavior.

1. Intentionally tapping my brakes when another car follows too closely
2. Making rude gestures at other drivers when they do something I do not like
3. Honking when another driver does something inappropriate
4. Merging into traffic even when another driver tries to close the gap between cars
5. Speeding up when another car tries to overtake me
6. Following another car in front of me closely to prevent another car from merging in front of me
7. Flashing my high beams at slower traffic so that it will get out of my way
8. Following a slower car at less than a car’s length
9. Driving 20 miles per hour above the speed limit
10. Passing in front of a car at less than a car’s length
11. Accelerating into an intersection when the traffic light is changing from yellow to red
12. Shouting loudly when another driver does something inappropriate
13. Driving while using a cellular phone, smartphone, etc.
14. At a “stop”, slowing down but not coming to a complete stop
15. Turning right and left without slowing down at intersections

Data from Houston et al., 2003, partially modified. Adapted with permission from Ref. [10]. 2003, Houston, J.M.

**Table 2 healthcare-10-02221-t002:** Profile of the participants.

	High Moral Group Mean (SD)	Low Moral Group Mean (SD)	*p*-Value
Male/Female (n) Age (years) Driving experience (years)	5/4	5/4	n.s.
	23.9 (5.7)	25.9 (7.0)	n.s.
	5.4 (6.0)	7.4 (7.2)	n.s.

Statistical analysis: The Mann–Whitney U test was performed; n.s.: not significant.

## Data Availability

The data presented in this study are available upon request from the corresponding author. The data are not publicly available because of privacy concerns.

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
