# Peer review of "Relationship between Moral Values for Driving Behavior and Brain Activity: An NIRS Study"

_healthcare, 2022, doi:10.3390/healthcare10112221_

Round 1
Reviewer 1 Report
I read this study with great interest and found it very useful especially for colleagues studying risky driving behaviors. The innovative effort of this work is the use of methods such as NIRS which assess brain activity. Specifically, the authors have explored “the moral view of driving behavior in healthy drivers” with the use of near-infrared spectroscopy in order to assess the “brain activity while eliciting moral emotions” to the participants. As far as I have understood, to achieve this task, authors have used two groups, “one with high moral values and the other with low moral values”, and have looked for significant differences in “brain activity between these groups”.
I believe this study should be published in your journal after the authors taking into consideration some minor modifications.
The term "dangerous" could be replaced by the more appropriate term "risky", eg., risky driving behaviour instead of dangerous.
Introduction
Please verify that "the third leading cause of death for young people aged 15-29 years old worldwide" are road traffic injuries.
"However, there are drivers who, for one reason or another, engage in self-centered and dangerous driving behavior and ignore traffic norms". (Please insert some references).
In addition, they may also become angry at the words and actions of others, resulting in aggressive and dangerous driving behavior (references pls, e.g. Journal of Safety Research Volume 33, Issue 4, 1 December 2002, Pages 431-443).
"...Additionally, drivers with a low sense of legal obligation are believed to disregard traffic rules and have no guilt... [5]",(more ref, pls).
I suggest to extend your literature review and insert some more references which directly refer, with or without using NIRS, to the aim of your manuscript.
To have a more elegant presentation of the paper I suggest introducing a hypothesis and 1-2 research questions which will be controlled and discussed.
Finally, I suggest introducing the weaknesses and strengths of your study.
Reviewer 2 Report
The main aim of this study was to examine the relationship between moral values for driving behavior and 66 brain activity using NIRS. Authors used questionnaires and brain activity measurement to investigate the connection between moral values and brain activity. The manuscript is in general well written, but I’m missing the elaboration on why such study is necessary. Also, some important methodological descriptions are missing. Moreover, the study included low number of participants.
More detailed comments are presented bellow:
1. I would rename subsection 2.3 into “Research procedure” since it describes testing procedure. In that subsection authors could move the sentence from row 71.
2. Regarding research procedure, can authors explain what instructions they gave to participants. Did participants knew what is the goal of a study? What was the setting in the testing room (were there any “distractions” which could impacted results)?
3. Is 20 s pause enough? It could have easily been that participants were still “affected” by the last test block when new one started. The same applies for sentences in one block, and again is 10 s enough?
4. I’m concern about the sample size which is really low and from it we cannot gain some solid generalized conclusions. How do authors comment on that?
5. Figure 3 is poor quality and it is hard to see what is in the charts.
6. In the Discussion authors should elaborate limitation of the study.
7. What is the practical value of the findings (except for opening up a discussion). Is there a way that we can use obtained results in more practical sense?
Round 2
Reviewer 2 Report
Dear author,
Authors have addressed all my comments so from my point of view, manuscript can be accepted.